First records of extinct kentriodontid and squalodelphinid dolphins from the Upper Marine Molasse (Burdigalian age) of Switzerland and a reappraisal of the Swiss cetacean fauna

Aguirre-Fernández Gabriel gabriel.aguirre@pim.uzh.ch 1
Jost Jürg 2
Hilfiker Sarah 1 3
1 Paleontological Institute and Museum, University of Zurich , Zurich , Switzerland
2 Zofingen , Switzerland
3 Current affiliation:  Department of Environmental Systems Science, Swiss Federal Institute of Technology , Zurich , Switzerland
Hedrick Brandon
Electronic publication date: 2022 May 16
Publication date: 2022
Volume: 10
Electronic Location ID: e13251
Received 2021 Oct 1; Accepted 2022 Mar 21
Copyright: ©2022 Aguirre-Fernández et al.
Copyright year: 2022
Copyright holder: Aguirre-Fernández et al.
License: This is an open access article distributed under the terms of the Creative Commons Attribution License, which permits unrestricted use, distribution, reproduction and adaptation in any medium and for any purpose provided that it is properly attributed. For attribution, the original author(s), title, publication source (PeerJ) and either DOI or URL of the article must be cited.
License URL: https://creativecommons.org/licenses/by/4.0/

Keywords: Cetacea, Odontoceti, Burdigalian, Upper Marine Molasse, Periotic, Paratethys, Kentriodontidae, Squalodelphinidae, Physeteridae, Kentriodon

Funding: Fonds für Lehre und Forschung (Natural History Museum Basel) The Synthesys Programme (BE-TAF project 4644) This project was funded by Fonds für Lehre und Forschung (Natural History Museum Basel) and the Synthesys Programme (BE-TAF project 4644). The research group led by Marcelo R. Sánchez-Villagra also provided financial support. The funders had no role in study design, data collection and analysis, decision to publish, or preparation of the manuscript.

==============================
The Swiss Upper Marine Molasse (OMM) documents a transgression event dated to around 21 to 17 million years in which dolphin and other vertebrate remains have been reported. We revised the whole cetacean (whales and dolphins) OMM assemblage available in main collections, focusing on the identification and interpretation of periotics (bone that contains the inner ear). Periotics are rare, but they provide the richest taxonomic information in the sample and hint to environmental associations. Micro-computerized tomography allowed the reconstruction of bony labyrinths for comparisons and environmental interpretations. Three families are represented by periotics: Kentriodontidae, Squalodelphinidae and Physeteridae. The cetacean taxonomic composition of the Swiss OMM reinforces biogeographical patterns reported for the Mediterranean and Paratethys during the Burdigalian at a regional scale and the Calvert cetacean fauna of the northwest Atlantic at oceanic scale.

Introduction

The Swiss Molasse (Fig. 1) is a textbook example of a foreland basin (Sissingh, 1998) and includes two marine transgression-regression cycles, spanning from the Rupelian to the Serravalian (Labhart, 1985; Swiss Committee on Stratigraphy, 2020). Among the vertebrate fossils of the Molasse, remains of cetaceans (whales and dolphins) are known. Climatic changes and major geographic rearrangements in the Tethys and Paratethys (including the closure of the Tethys Seaway) may have played a significant role in the cetacean composition at regional and global scales (Steeman et al., 2009; Bianucci & Landini, 2002). Cetacean fossils in Swiss localities belong to sediments of the second transgression event, dated 21–17 million years ago (Ma). The Upper Marine Molasse or ‘Obere Meeresmolasse’ (hereafter referred to as OMM) is an informal lithostratigrahic group composed of two formations; the underlying Lucerne Formation, and the St. Gallen Formation. Jost, Kempf & Kälin (2016) provided a comprehensive stratigraphic overview and discussed the palaeocological interpretations.

Figure 1 Reconstruction of the Paratethys during the Burdigalian.

(A) Western Europe with Switzerland marked in red, modified from Rögl (1998) and Berger et al. (2005); (B) maximal flooding of the Paratethys over the Swiss Plateau at ca. 18–17.5 Ma, during the deposition of the St. Gallen Fm., based on Schlüchter et al. (2019). Localities: Brüttelen (orange), Madiswil (yellow), and Staffelbach (green).

The OMM cetacean assemblage is represented by a large, but fragmentary sample. Similar preservation patterns are known for other coeval localities such as the ‘Molasse of Baltringen’ in Germany and ‘Pietra di Cantoni’ in northern Italy (Bianucci & Landini, 2002). Here, we focus on the description of seven well-preserved periotics and revise all (new and previously reported) cetacean remains in major Swiss collections in order to provide an overview in the context of new taxonomic advances.

The periotic bone contains the inner ear (cochlea and semicircular canals) and has become isolated from the skull in many odontocetes (Mead & Fordyce, 2009). This element not only provides substantial taxonomic information, but also insights into habitat preferences (Costeur et al., 2018) and is therefore extremely valuable in highly-fragmentary assemblages (Aguirre-Fernández et al., 2017; Steeman, 2009).

The Miocene fossil record of cetaceans (whales and dolphins) in the circum-Mediterranean region (Mediterranean and Paratethys) is known from localities in Egypt, southern France, southern Germany, Italy, Malta, Spain and Switzerland (Bianucci et al., 2011; Dominici et al., 2020). Revisions of several neighbouring faunas prompted this reappraisal of Swiss specimens. The current work builds upon the overview by Pilleri (1986a). We report hitherto unknown kentriodontid and squalodelphinid fossils and we dispute the presence of putative delphinids in the Swiss Marine Molasse. This paper takes a more conservative view on taxonomic affinities than that in Pilleri (1986a). Figure 2 shows the relationships of families reported as present in the Mediterranean and Paratethys during the Burdigalian in Bianucci & Landini (2002) and in the OMM according to Pilleri (1986a).

Figure 2 Families present in the Mediterranean and Paratethys during the Burdigalian according to Bianucci & Landini (2002).

Topology based on Lloyd & Slater (2021), except for Dalpiazinidae (nomen dubium), for which the hypothesis by Muizon (1988a) is depicted. Groups marked with an asterisk were reported as present in the OMM by Pilleri (1986a). Delphinids (gray) were reported in Pilleri (1986a) for the OMM, but here reassigned to Kentriodontidae. Silhouettes represent groups with extant representatives and are credited to Chris Huh (CC BY-SA 3.0), obtained from phylopic.org.

Materials and Methods

Anatomical descriptions follow the nomenclature of Mead & Fordyce (2009) for external (bone) structures and Ekdale (2013) for internal (bony labyrinth, i.e., cochlea and semicircular canals) structures. External measurements of the periotic were made with calipers, following Kasuya (1973). Open nomenclature follows the recommendations in Sigovini, Keppel & Tagliapietra (2016). Micro-computed tomography (µCT) data of seven periotics were obtained at the University of Zurich using a Nikon XT H 255 ST µCT scanner (scanning resolution of 20 µm). Segmentation of the earbones and their bony labyrinth endocasts was performed using Mimics Innovation Suite 19.0. Bony labyrinth measurements that serve as correlates for hearing sensitivity are based on the methods described in Racicot & Preucil (2021). The 3D models of all the periotics and their bony labyrinths are available at the MorphoMuseuM repository (Aguirre-Fernández, Hilfiker & Jost, 2022). A phylogenetic analysis based on the matrix in Kimura (2018) and originally published in Lambert, Bianucci & Urbina (2014, see their appendix 1 for list of characters) was performed. The matrix included 23 taxa and 37 characters, for which only the periotic characters were coded for NMBE 5023942 (characters 20–26: 1020000) and NMBE 5023943(characters 20–26: ?120000), leaving all other characters as “?”. The parsimony analysis was performed in TNT 1.5 (Goloboff & Catalano, 2016), treating all characters as unordered (non-additive) with equal weights. The search was performed under the default settings under “traditional search” (TBR swapping algorithm, 10 trees were saved per replication) for 100,000 replications.

Collection acronyms

MGL Cantonal Museum of Geology Lausanne, Lausanne, Switzerland.

NHMB Natural History Museum Basel, Basel, Switzerland.

NMBE Natural History Museum Bern, Bern, Switzerland.

PIMUZ Paleontological Institute and Museum, University of Zurich, Zurich, Switzerland.

Results

General remarks on OMM cetaceans

A total of 290 cetacean elements were observed in the collections housed at MGL, NHMB, NMBE and PIMUZ. All elements were found isolated (i.e., single bones rather than articulated skeletons); almost all were fragmentary (i.e., bones were broken and had missing parts) and some were also abraded or polished. The most frequent elements were by far teeth (69%), followed by vertebrae (16%) and periotics (7%). Teeth and vertebrae are of poor taxonomic value and belong to a range of odontocete groups. Few other skull elements are also too fragmentary for unambiguous identification. The Table S1 includes basic information on all material studied, with previous (i.e., Pilleri, 1986a) and new (this study) taxonomic opinions. We focus below on the periotics and their bony labyrinths because they are the most informative elements for taxonomy and environmental interpretations available in the OMM cetacean sample.

Systematic Palaeontology

Cetacea Brisson, 1762	
Odontoceti Flower, 1864	
Delphinida de Muizon, 1988b	
Kentriodontidae Slijper, 1936	
cf. Kentriodon	
(Figs. 3–4)	

Description and remarks

NMBE 5023944 (left periotic), figured in Pilleri (1986a:Plate 5D); NMBE 502345 (right periotic) figured in Pilleri (1986a:Plate 5G); NMBE 5023946 (left periotic) mentioned in Pilleri (1986a: p.29), all three from Brüttelen-Fluh; and NMBE 5036436 (right periotic; figured in Pilleri (1986a:Plate 8K) from Madiswil-Ghürn. All four specimens were identified as delphinidans in Pilleri (1986a). Both localities correspond to sediments of the Lucerne Fm. (Fig. 1). The Swiss kentriodontid periotics strongly resemble several species of Kentriodon, such as K. pernix, K. obscurus, K. hoepfneri, K. nakajimai and K. sugawarai in overall shape, dimensions and proportions. Small variation in shape (e.g., in the pinching of the anteroventral angle or the ventral inflexion of the posterior process of the periotic), size and proportions observed among the Swiss kentriodontid periotics (Fig. 3) are within the range of intraspecific variation reported by Barnes & Mitchell (1984) for a sample of 31 isolated periotics from the Sharktooth Hill Bonebed and may therefore represent a single species. The anterior process of kentriodontids and other delphinidans is short, thick and with a squared-off (dorsoventral) margin in lateral view (Kasuya, 1973). The apex of the anterior process is mediolaterally pinched and slightly deflected medially. The length of the anterior process is similar to that of the pars cochlearis (Table 1); the posterior process is relatively short (anteromedially) and directed ventrally; the outline of the pars cochlearis is slightly oval, longer (in anteroposterior axis) than it is wide. The aperture for the cochlear aqueduct is located dorsally and posterior to the aperture for the vestibular aqueduct, both aqueducts are roughly the same size (Fig. 3). The mallear fossa is round; the vestibular window is round and relatively large; the fenestra rotunda is teardrop-shaped; the posterior bullar facet is smooth (Fig. 3). There is an anterointernal sulcus clearly visible in medial view (Fig. 3). The parabullary ridge is ventrally concave (Fig. 3).

Figure 3 Periotics of cf. Kentriodon.

(A–D) NMBE 5023944; (E–H) NMBE 5023945; (I–L) NMBE 5023946; and (M–P) NMBE 5036436 featuring the 3D models (available for download from the MorphoMuseuM repository). The lower row illustrates anatomical landmarks of the periotic as seen in NMBE 5036436. Views: dorsal: A, E, I, M; ventral: B, F, J, N; medial: C, G, K, O; lateral: D, H, L, P.

Figure 4 Bony labyrinths of cf. Kentriodon.

(A–C) NMBE 5023944; (D–F) NMBE 5023945 (reflected); (G–I) NMBE 5023946; (J–L) NMBE 5036436 (reflected). The lower row illustrates anatomical landmarks of the bony labyrinth as seen in NMBE 5036436. Views: anterior: A, D, G, J; dorsal: B, E, H, K; lateral: C, F, I, L.

Table 1 Periotic and inner ear measurements of the fossil kentriodontids (in mm); 1 USNM 8060 as externally measured by Kellogg (1927a) and internally (bony labyrinth) by Churchill et al. (2016); + as preserved.

	NMBE	NMBE	NMBE	NMBE	Kentriodon	
	5023944	5023945	5023946	5036436	pernix 1	
Periotic						
Greatest length of periotic	25	+ 25	27	23.6	28.8	
Width of the periotic	16.6	15.3	15.1	14.5	16.9	
Length of pars cochlearis	12	12.6	13.3	13.7	—	
Height of pars cochlearis	8.1	9	9.3	7.9	10.5	
Width of the pars cochlearis	8	8.3	7.7	8.5	—	
Length of anterior process	12	12	11.3	13	13.4	
Inner ear endocast						
Cochlear turns (t)	1.5	1.7	1.5	1.7	1.7	
Cochlear length	25	29	29	27	27.6	
Axial height (h)	4	4	4	3.7	4	
Axial pitch (h/t)	2.7	2.3	2.7	2.1	2.3	

The bony labyrinths of NMBE 5023944–5023946, and NMBE 5036436 (Fig. 4) share features of other odontocetes, such as the small vestibular apparatus as compared to the cochlea, the low number of spiral turns in the cochlea and their loose coiling. Although the comparisons are limited because the bony labyrinth of Kentriodon pernix remains undescribed, published cochlear measurements of the bony labyrinth of Kentriodon pernix indicate a strong similarity to the Swiss kentriodontids (Table 1). PlatanistoideaGray, 1863	
Squalodelphinidae fam. gen. sp.Dal Piaz, 1917	
(Figs. 5–6)	

Description and remarks

NMBE 5023942 (right periotic), figured in Pilleri (1986a: Plate 5F) and NMBE 5023943(left periotic), figured in Pilleri (1986a: Plate 5E) were both found in Brüttelen-Fluh (Lucerne Fm). Both periotics were identified in Pilleri (1986a) as squalodontids. Our phylogenetic analysis returned 216 equally-parsimonious trees with a score of 59. The majority rule tree is very similar to that of Lambert, Bianucci & Urbina (2014) and Kimura (2018) and places NMBE 5023942 and NMBE 5023943 as two distinct taxa within a clade formed by Squalodelphinidae + Dilophodelphis (Fig. 7). The synapomorphies of the clade that includes NMBE 5023942 and sister taxa is the presence of an articular rim—labeled in Fig. 5 as recurved lateral sulcus based on Aguirre-Fernández & Fordyce (2014); and a large aperture of the cochlear aqueduct (characters 20 and 22 of Lambert, Bianucci & Urbina, 2014). The OMM earbones (Fig. 5) differ in the following features: (1) the shape of their anterior processes (being longer and more slender in NMBE 5023943), (2) the prominent anterointernal sulcus of NMBE 5023943 (absent in NMBE 5023942, (3) the ventral deflection of the anterior process in NMBE 5023942 (absent in NMBE 5023943); and (4) the deeper anterior bullar facet in NMBE 5023942. The prominent anterior bullar facet with well-defined medial and lateral boundaries (sensu Lambert, Bianucci & Urbina, 2014, Fig. 6) is a diagnostic character also present in other squalodelphinids such as Squalodelphis fabianii, Notocetus vanbenedeni, and Huaridelphis raimondii. The tuberosity in the posteromedial part of the anterior process is also present in Huaridelphis raimondii (see Lambert, Bianucci & Urbina, 2014, figs. 6A and B) and other squalodelphinids (e.g., Squalodelphis fabianii and Notocetus vanbenedeni), but is not restricted to this group, as it is also shown in some squalodontids, eurhinodelphinids, xenorophids and Waipatia (Lambert, Bianucci & Urbina, 2014; Fordyce, 1994). Some putative family-diagnostic characters such as a square-shaped pars cochlearis and a dorsally-oriented aperture for the cochlear aqueduct (sensu Lambert, Bianucci & Urbina, 2014) are absent in NMBE 5023942: the pars cochlearis has a relatively circular outline, the aperture for the cochlear aqueduct is indeed large, but not dorsally-oriented. The Swiss squalodelphinid periotics are smaller than Phocageneus, and comparable in size to Huaridelphis raimondii, the smallest known member of Squalodelphinidae (Lambert, Bianucci & Urbina, 2014).

Figure 5 Periotics of Squalodelphinidae indet.

(A–D) NMBE 5023942; (E–H) NMBE 5023943. Views: anterior: A, E; ventral: B, F; medial: C, G; lateral: D, H.

Figure 6 Bony labyrinths of Squalodelphinidae indet.

(A–C) NMBE 5023942 (reflected horizontally); (D–F) NMBE 5023943. Views: anterior: A, D; dorsal: B, E; lateral: C, F.

Figure 7 Fifty-percent majority rule tree summarizing the 216 equally parsimonious trees obtained from the parsimony analysis.

Allodelphinidae and Eurhinodelphinidae collapsed. Numbers below nodes indicate their frequency among trees in percentage.

The bony labyrinths of NMBE 5023942 and 5023943 are shown in Fig. 6. The shape of the cochlea is relatively flat compared to other platanistoids such as Waipatia and Awamokoa (for comparisons, see Viglino et al., 2021). Published measurements of the bony labyrinths of Phocageneus and Notocetus vanbenedeni show a slightly larger cochlear length and a quarter to half a cochlear turn more than the OMM squalodelphinids, but the axial pitches are overall very similar (Table 2).

Physeteroidea Gray, 1821	
Physteridae Gray, 1821	
Physeteridae indet.	
(Figs. 8–9)	

Description and remarks

NMBE 5036437 (left periotic) was found in Staffelbach-Böl (St. Gallen Formation). The periotic is comparatively large and robust (Fig. 8). Of the four characters relevant to the periotic mentioned in the phylogenetic analysis of Lambert, Bianucci & De Muizon (2017), NMBE 5036437 shares with other physeteroids the very small anterior bullar facet and the enlarged accessory ossicle (judged by the size of the fovea epitubaria). The accessory ossicle is fused to the periotic in some physeteroids (e.g., the Gross Pampau physeteroid in Montañez Rivera & Hampe, 2020), but not in NMBE 5036437 (accessory ossicle missing). The posterior part of the posterior process of NMBE 5036437 is directed posteroventrally as in other physeterids and unlike in kogiids. The high and small dorsal crest (lateral to the internal acoustic meatus) of NMBE 5036437 is a feature seen in other physeterids such as Aulophyseter, Orycterocetus and Physeter. NMBE 5036437 falls in the size range of both Aulophyseter and Orycterocetus, overall shape and proportions of the pars cochlearis and the anterior and posterior processes resemble Aulophyseter morricei Kellogg 1927b, but some features are also comparable to Orycterocetus crocodilinus Kellogg 1965 and deserve further comparisons, which were done using photos of the holotypes of Aulophyseter morricei and Orycterocetus crocodilinus, hereafter referred to by their generic names: In dorsal view, the pars cochlearis of NMBE 5036437 is larger than that of Aulophyseter and Orycterocetus, but closer in proportions to Aulophyseter. The elongated shape of the internal acoustic meatus resembles Orycterocetus. The aperture for the cochlear aqueduct is larger than the aperture of the vestibular aqueduct as in Orycterocetus. The anterior tip of the anterior process points anteriorly as in Orycterocetus. In ventral view, the fenestra rotunda has a kidney-shaped outline, which is distinct from both Aulophyseter and Orycterocetus. The anterior process is square-shaped and facing ventrally as in Aulophyseter. The posterior process is more slender than in both Aulophyseter and Orycterocetus, and the tip of the process is pointing slightly more ventrolaterally. The posterior bullar facet is smooth, unlike in both Aulophyseter and Orycterocetus, but it is unclear whether this is the result of abrasion. In medial view, the anterior process is more robust (higher), and the dorsal crest is less pronounced than in both Aulophyseter and Orycterocetus.

Table 2 Periotic and inner ear measurements of fossil squalodelphinids (in mm; e = estimated); 1 as externally measured by Kellogg (1957) for USNM 21039, and internally measured by Churchill et al. (2016) for USNM 182942; 2 as measured in Viglino et al. (2021).

	NMBE 5023942	NMBE 5023943	Phocageneus 1	Notocetus 2	
Periotic					
Greatest length of periotic	33(e)	35(e)	40	—	
Width of the periotic	19.8	15.9	20	—	
Length of pars cochlearis	13.9	14.7	—	—	
Height of pars cochlearis	11.4	10	14.2	—	
Width of the pars cochlearis	8.8	8.2	—	—	
Length of anterior process	15.6	16	21	—	
Inner ear endocast					
Cochlear turns (t)	1.5	1.7	2	2	
Cochlear length	26	30	43.5	32	
Axial height (h)	3.8	4	4.7	5.9	
Axial pitch (h/t)	2.5	2.3	2.3	2.9	

Figure 8 Periotic of Physeteridae indet.

NMBE 5036437. Views: dorsal: A; ventral: B; medial: C; lateral: D.

Figure 9 Bony labyrinth of Physeteridae indet.

NMBE 5036437. Views: anterior: A; dorsal: B; lateral: C.

Although the external dimensions of the periotic NMBE 5036437 are very similar to those of Aulophyseter morricei, there are strong differences in the cochlear length and axial height, also reflected in the axial pitch (Table 3 and Fig. 9).

Table 3 Periotic and inner ear measurements of fossil physeterid NMBE 5036437, and Aulophyseter morricei (in mm; e = estimated); 1average of up to 9 periotics, as externally measured in Kellogg (1927b, p.20) and internally (bony labyrinth) by Churchill et al. (2016, Table S2) for SDSNH 55015.

	NMBE 5036437	Aulophyseter morricei 1	
Periotic			
Greatest length of periotic	37.3	39	
Width of the periotic	25.7	26	
Length of pars cochlearis	21.6	—	
Height of pars cochlearis	21.3	19	
Width of the pars cochlearis	12	—	
Length of anterior process	20	20.1	
Inner ear endocast			
Cochlear turns (t)	1.7	1.7	
Cochlear length	43	32.1	
Axial height (h)	7.5	5.7	
Axial pitch (h/t)	4.3	3.2	

Discussion

At a larger scale, the connection of the Paratethys with the Indian Ocean and the Mediterranean during the Aquitanian favoured the distribution of warm-water faunas; these conditions prevailed until the late Burdigalian, when the seaway between the Mediterranean and the Indian Ocean closed, the eastern Paratethys became isolated (forming the so-called Kotsakhurian Sea) and the central/western Paratethys became much reduced (Rögl, 1998). The late Burdigalian is marked by a large diversity of odontocetes and the subsequent demise of many longirostrine forms, possibly linked to climatic changes at the beginning of the Middle Miocene (Bianucci & Landini, 2002). The kentriodontid and squalodelphinid periotics here reported come from localities of the Brüttelen-Muschelnagelfluh Member, right at the base of the Lucerne Formation and therefore from older sediments of the OMM. The Brüttelen-Muschelnagelfluh Member is chronologically interpreted at the base of the Burdigalian and environmentally interpreted as shallow marine (Schwab, 1960). In contrast, the physeterid periotic was found in sediments of the Staffelbach-Grobsandstein Bed, a local unit at the base of the St. Gallen Fm which is environmentally interpreted as sublittoral (ca. 100 m deep) based on the rich chondrichthyan composition (Jost, Kempf & Kälin, 2016). Further, the chondrichthyan composition of the Staffelbach-Grobsandstein Bed is extremely similar to that of the Rhone Valley (Jost, Kempf & Kälin, 2016), a pattern also reported for the mollusc and echinoid faunas, which place Switzerland in a transitional zone between the Central Paratethys faunas (eastwards) and the Rhone Basin and the Mediterranean faunas(westwards) for the Early Burdigalian (Kroh & Menkveld-Gfeller, 2006).

Despite recent efforts to disentangle the relationships of kentriodontids and redefine the group, their monophyly is still a matter of debate (e.g., Guo & Kohno, 2021; Peredo, Uhen & Nelson, 2018; Lambert et al., 2017). Regardless, the type-bearing genus Kentriodon and its closest relatives were cosmopolitan and diverse in the early Miocene (Guo & Kohno, 2021). Bianucci & Landini (2002) reported the presence of kentriodontids in five Burdigalian-Langhian European localities: Baltringen (southern Germany), Rosignano and Vignale (northern Italy), Cursi-Melpignano quarries of the Salento Peninsula (southern Italy), and Switzerland. The designation of NMBE 502344 and NMBE 502345 (Fig. 3) to Kentriodontidae corroborates the suggestion already made by Bianucci & Varola (1994), contrasting with a previous identification as delphinidan earbones (Pilleri, 1986a see plate 5 D & G and plate 8 K). The two skull-based and highly-diagnostic kentriodontid species Rudicetus squalodontoides (Burdigalian–Messinian, 18–6 Ma) and Tagicetus joneti (late Serravallian, 12.7–11.6 Ma) do not have preserved periotics (Bianucci, 2001; Lambert, Estevens & Smith, 2005). Bianucci & Varola (1994) reported kentriodontid periotics from the same area as R. squalodontoides (Pietra leccese), contemporaneous with the Swiss localities. Further, Bianucci & Varola (1994) reassigned other earbones (previously recognized as Delphinidae in Pilleri, 1986b; Pilleri, Gihr & Kraus, 1989) from Piedmont and Baltringen to Kentriodontidae. Kentriodon hoepfneri from Gross Pampau, Germany (Kazár & Hampe, 2014) and the kentriodontid remains from Bihor County, Romania (Kazár & Venczel, 2003) are from younger (middle Miocene) sediments. Studies on intraspecific variation of periotics are needed to better understand their disparity. Barnes & Mitchell (1984) interpreted a large sample of isolated periotics from the Sharktooth Hill Bonebed as belonging to a single species (Kentriodon obscurus), combining two species previously known as Grypolithax obscura and Grypolithax pavida, both described in Kellogg (1931). Remarkably, Barnes & Mitchell (1984) listed six characters that denote the range of intraspecific variation within Kentriodon obscurus, whereas the only noticeable morphological difference separating the holotype of Kentriodon pernix from this sample was the more circular internal acoustic meatus (Kentriodon pernix was reported from the Calvert Formation (North Atlantic), whereas the Sharktooh Hill Bonebed is located in the North Pacific). Kasuya (1973) reported little intraspecific variation in extant species, as Martins et al. (2020) also did for the cochleae of harbor porpoises. Bony labyrinth measurements that correlate to hearing sensitivity indicate that Kentriodon pernix (and possibly other kentriodontids) may have been among the earliest odontocetes to use a narrow-band-high-frequency (NBHF) biosonar (Racicot & Preucil, 2021; Galatius et al., 2018). NBHF may have evolved to avoid predation by large echolocating predators such as orcas (Morisaka & Connor, 2007), but their fossil record does not extend into the Miocene. Odontocetes with a similar niche to orcas include macroraptorial physeteroids (Racicot & Preucil, 2021; Galatius et al., 2018) and ‘squalodontids’ (Kellogg, 1923), both abundant in European Miocene localities.

Squalodelphinidae is a monophyletic group sister to Platanistidae (Lambert, Bianucci & Urbina, 2014) with a distribution in both latitudinal hemispheres in the Pacific and Atlantic coasts (Bianucci, Urbina & Lambert, 2015). The highest diversity centers in the North Atlantic at around the early Miocene and suggests a close connection between the European and North American faunas (Bianucci, Urbina & Lambert, 2015). The periotics here described represent the first record of Squalodelphinidae in Switzerland and are contemporaneous with Medocinia tetragorhina from the Burdigalian locality Saint-Medard-en-Jalle in France (de Muizon, 1988a) and Squalodelphis fabianii from the Libano Sandstone in northern Italy (Bianucci & Landini, 2002; Dal Piaz, 1917), of which the periotics are either lacking (in the former) or unprepared and still in situ (probably in the latter, as the tympanic bulla is still in situ). Smaller squalodelphinids such as the OMM specimens, about the size of Huaridelphis raimondii could be interpreted as having occupied a similar niche to that of the extant Delphinus delphis, preying on small fish (Bianucci et al., 2018). The identification of NMBE 5023942 and NMBE 5023943 as squalodelphinids remains tentative, as eurhinodelphinids also show a similar morphology, including a developed anterior bullar facet and a large aperture for the cochlear aqueduct. The pars cochlearis is relatively round (particularly in NMBE 5023942) and not square-shaped as in many squalodelphinids.

Among the taxa here studied, Physeteridae is also attested by the many teeth from the OMM housed in collections, as already reported in Pilleri (1986a). Here, the physeteroid Helvicetus rugosus Pilleri 1986a is regarded as nomen dubium. The range of sizes, shapes and degrees of wear of physeteroid teeth suggest a high diversity of this group in the OMM, but a revision of the teeth is out of the scope of this paper. Bianucci & Landini (2002) reported the presence of physeterids in many Burdigalian-Langhian localities around the Mediterranean (Baltringen, the Rhone Valley, Rosignano and Vignale, and the Salento Peninsula) ranging all across the Miocene, pointing to a considerable radiation of this group in the area at that time. Several isolated physeterid periotics are known from the ‘pietra leccese’ (Salento Peninsula) and ‘pietra di cantoni’ (Rosignano and Vignale), possibly representing more than five genera, according to Bianucci & Landini (2002).

Pilleri (1986a) mentioned that Cuvier reported a scapula that can only belong to Balaenoptera, reportedly found in Lake Geneva, but such a specimen was not located. Given the age of the sediments and the size of the isolated elements that can only be diagnosed to Cetacea indet., we assume that only odontocetes are represented in the sample. This pattern reflects a global early Miocene ‘dark age’ for mysticetes, which has been linked to environmental changes at around the Oligocene-Miocene boundary and led to the decline of coastal assemblages. While toothed mysticetes went extinct, filter feeders thrived offshore and recolonized coastal environments in the middle Miocene (Marx, Fitzgerald & Fordyce, 2019).

Overall, the faunal composition of the OMM fits the interpretations outlined in Bianucci & Landini (2002) for the Mediterranean/Paratethys fauna during the Burdigalian. On a broader geographic scale, there is a clear association with the contemporaneous and extremely diverse Calvert fauna on the eastern coast of North America, with representatives of at least six families in common: Squalodontidae, Eurhinodelphinidae, Squalodelphinidae, Kentriodontidae, Physeteridae and Ziphiidae (Bianucci & Landini, 2002; Gottfried, Bohaska & Whitmore Jr, 1994).

Conclusions

There is a prevalence of isolated, fragmented, and sometimes abraded cetacean remains in the OMM. The teeth are the most frequent elements. Periotics are rare, but diagnostic. The seven periotics herein described attest to the presence of Kentriodontidae, Squalodelphinidae (two morphotypes) and Physeteridae. Previous assignations of periotics to Delphinidae in Pilleri (1986a) plate 5 D & G and plate 8 K in the OMM (and elsewhere; see Bianucci & Varola, 1994) are indeed kentriodontids. Previous assignations of periotics to Squalodontidae Pilleri (1986a) plate 5 E & F in the OMM are here identified as squalodelphinids. Physeteridae is represented by one periotic from the St. Gallen Formation. The faunal composition is similar to that reported for the Burdigalian at a regional (Mediterranean and Paratethys) scale (Bianucci & Landini, 2002), with representatives of families also found in the Calvert Fm on the western Atlantic coast (Gottfried, Bohaska & Whitmore Jr, 1994).

Supplemental Information

Supplemental Information 1 Overview on all elements studied from main Swiss collections

Click here for additional data file.

We thank the academic editor Brandon Hedrick and reviewers Rachel Racicot, Toshiyuki Kimura and Robert Bossenecker for their comments. Loïc Costeur (NHMB), Olivier Lambert (RBINS), Ursula Menkveld (NMBE), Antoine Pictet (GML) and Christian Klug (UZH) are thanked for access to material. Beat Lüdi is thanked for allowing the study of and donating the physeterid to the NMBE. Loïc Costeur (NHMB) kindly scanned specimen NMBE 5036436. Christian Meyer, Martin Schneider and Markus Weick are also thanked for their help while at NHMB. Bernhard Hostettler is thanked for help while at NMBE. O. Lambert is thanked for discussions and access to specimen photos for comparisons. Thomas Schmelzle is thanked for discussions on inner ear anatomy. Dylan Bastiaans is thanked for advice and help with segmentation. Marcelo R. Sánchez-Villagra, Rachel Racicot and Aldo Benites are thanked for comments on an earlier draft. We thank the Willi Hennig Society for making TNT available to anyone interested in using this software.

Additional Information and Declarations

Competing Interests

Author Contributions

Animal Ethics

Data Availability

The authors declare there are no competing interests.

Gabriel Aguirre-Fernández conceived and designed the experiments, performed the experiments, analyzed the data, prepared figures and/or tables, authored or reviewed drafts of the paper, and approved the final draft.

Jürg Jost analyzed the data, authored or reviewed drafts of the paper, and approved the final draft.

Sarah Hilfiker performed the experiments, analyzed the data, prepared figures and/or tables, and approved the final draft.

The following information was supplied relating to ethical approvals (i.e., approving body and any reference numbers):

The vertebrates described in this study are fossils.

The following information was supplied regarding data availability:

An overview on all the cetacean fossils from the Upper Marine Molasse is available in the Supplementary File.

The 3D models of periotics and their bony labyrinths are available at MorphoMuseum.

Aguirre-Fernández G., Jost J., Hilfiker S., 2022. 3D models related to the publication: First records of extinct kentriodontid and squalodelphinid dolphins from the Upper Marine Molasse (Burdigalian age) of Switzerland and a reappraisal of the Swiss cetacean fauna. MorphoMuseuM 8:159. doi: 10.18563/journal.m3.159

NMBE 5023942: Aguirre-Fernández G., Jost J., Hilfiker S., 2022. M3#862_NMBE 5023942. doi: 10.18563/m3.sf.862

NMBE 5023943: Aguirre-Fernández G., Jost J., Hilfiker S., 2022. M3#863_NMBE 5023943. doi: 10.18563/m3.sf.863

NMBE 5023944: Aguirre-Fernández G., Jost J., Hilfiker S., 2022. M3#858_NMBE 5023944. doi: 10.18563/m3.sf.858

NMBE 5023945: Aguirre-Fernández G., Jost J., Hilfiker S., 2022. M3#859_NMBE 5023945. doi: 10.18563/m3.sf.859

NMBE 5023946: Aguirre-Fernández G., Jost J., Hilfiker S., 2022. M3#860_NMBE 5023946. doi: 10.18563/m3.sf.860

NMBE 5036436: Aguirre-Fernández G., Jost J., Hilfiker S., 2022. M3#861_NMBE 5036436. doi: 10.18563/m3.sf.861

NMBE 5036437: Aguirre-Fernández G., Jost J., Hilfiker S., 2022. M3#864_NMBE 5036437. doi: 10.18563/m3.sf.864.

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
