# Peer review of "First records of extinct kentriodontid and squalodelphinid dolphins from the Upper Marine Molasse (Burdigalian age) of Switzerland and a reappraisal of the Swiss cetacean fauna"

_PeerJ, doi:10.7717/peerj.13251_

## Round 0.1 · original submission · Minor Revisions

Dear authors,

Thank you for your submission to PeerJ. This is an interesting addition to the paleocetacean literature and the inclusion of uCT scans for inner ears of a variety of species is commendable. Three reviewers all suggested minor revisions prior to publication.

Most notably, two reviewers suggested you check the phylogenetic characters and affinities for squalodelphinids, which might require some reassessment of your specimens. Another reviewer suggested an introductory phylogenetic tree figure. I think that this would be immensely useful. I am not an expert on extinct cetaceans and I think it would have helped me as I read through the manuscript.

In addition to reviewer comments, there were a few grammar issues that will need revision:

Line 14: million should be lowercase

Line 41–42: Perhaps ‘and is therefore extremely valuable…’

Line 53: ‘with calipers’

Line 54: CT should be capitalized.

Line 68: ‘fragmentary’ rather than fragmented

Line 70: The word periotic is repeated

Line 189: Put Guo and Kohno 2021 in parentheses. Also line 194 with Pilleri

When you submit your revised version, please include a clean version of the manuscript, a tracked changes version, and a reviewer response document outlining how you responded to each reviewer comment. Please let me know if you have any questions.

Best,

Brandon P. Hedrick, Ph.D.

·

Basic reporting

The manuscript "First records of extinct kentriodontid and squalodelphinid dolphins from the Upper Marine Molasse (Burdigalian age) of Switzerland and a reappraisal of the Swiss cetacean fauna" by Aguirre-Fernandez et al. represents an interesting addition to our knowledge of the fossil record of toothed whale in the circum-Mediterranean region. The authors described seven periotics from the Swiss Upper Marine Molasse and revised the previous taxonomic assignment of the specimens. The description is based not only on the external characters but also on the internal characters by using micro CT scan. I suspect that this paper will be widely received by people interested in the evolution of cetaceans.

Experimental design

No comments

Validity of the findings

No comments

Additional comments

I have some comments that I think would improve the paper, but none of them are critical. I annotated the pdf file of the manuscript for the revision. Please see attached file of the annotated manuscript.

My main concern is on the following two points:
1) Lines 115-134: Taxonomical assignment to the Squalodelphinidae
The authors mentioned that "The prominent anterior bullar facet with well-defined medial and lateral boundaries is a diagnostic character also present in other squalodelphinids" (lines 121-122). But, this morphological character is also found in other group of odontocetes (e.g., eurhinodelphinids). The authors also mentioned that "The tuberosity in the posteromedial part of the anterior process is also present in Huaridelphis raimondii....and other squalodelphinids" (lines 123-126). But, as mentioned by the authors, the character is also found in another group of odontocetes. The authors also mentioned that "Some putative family-diagnostic characters...are absent in NMBE5023942" (lines 127-129). So, the reason why the authors diagnose the specimens as Squalodelphinidae remains unclear.

Please state on the characters suggesting that the specimens belong to the Squalodelphinidae. Perhaps, previous papers (e.g., Muizon, 1987; Kimura, 2018) discussing on the combination of the morphological characters in the periotic of the Squalodelphinidae might be help.

Muizon, C. de. 1987. The affinities of Notocetus vanbenedeni, an Early Miocene Platanistoid (Cetacea, Mammalia) from Patagonia, Southern Argentina. Novitates 2904:1-27.
Kimura, T. (2018). First squalodelphinid from the early Miocene of the Pacific realm in the Northern Hemisphere. Journal of Vertebrate Paleontology, 38(4). DOI: 10.1080/02724634.2018.1493595.


2) Lines 90-94 and 202-205: Intraspecific variation of the periotic in Kentriodon
Barnes and Mitchel (1984) discussed on the intraspecific variation of the periotic in Kentriodon based on a large number of the periotic from the Sharktooth Hill Bonebed. The paper might provide an insight on the interpretation of the morphological variations found in the studied specimens.

Barnes, L. G., and E. Mitchell. 1984. Kentriodon obscurus (Kellogg, 1931), A fossil dolphin (Mammalia: Kentriodontidae) from the Miocene Sharktooth Hill Bonebed in California. Natural History Museum of Los Angeles County, Contribution in Science, 353:1-23.

·

Basic reporting

No comment.

Experimental design

No comment.

Validity of the findings

No comment.

Additional comments

This interesting paper describes and places into geologic context the cetacean material available from several Swiss collections. It has the added bonus of measurements and descriptions based on CT scans and extracted cochleae of the periotic material, which contributes additional important natural history data for the specimens. Overall this paper is great as is, and I only have minor comments and edits that can be seen in the attached annotated PDF. I do think an introductory figure with a basic composite phylogeny containing the cetacean groups that are covered would place the work into even better evolutionary context for readers unfamiliar with extinct cetacean phylogenetic relationships, but it's not necessary.

With best regards,
Dr. Rachel Racicot

·

Basic reporting

The paper is very well written.

Experimental design

See below.

Validity of the findings

Acceptable; though I do have some questions about some of the identifications. See below.

Additional comments

This new study by Aguirre-Fernandez et al. re-describes a modest assemblage of cetacean earbones from the late early Miocene Upper Marine Molasse of Switzerland. This assemblage was previously reported by Pilleri (1986A/B). Pilleri was an enthusiastic and prolific cetologist but contemporary paleocetology has developed more refined standards for descriptions and figures, leaving many of his studies wanting. Therefore, contemporary redescription of this assemblage with updated figures is most welcome. In addition to reporting the external morphology of these earbones, the authors also CT-scanned the bony labyrinths of these specimens. This is remarkable, as this is the first time CT analysis of periotics has been reported for an entire odontocete assemblage; typically analyses focus on the adaptations and ecology of an individual species or are included in a wider morphofunctional analysis of hearing/balance adaptations across cetaceans. This assemblage based approach is novel and provides further ecological insight into Burdigalian-age cetaceans from the molasse of Switzerland. Altogether the paper is a competent advance in the study of Miocene cetaceans and I recommend minor revision. I have made some critical suggestions below. Sincerely, Robert W. Boessenecker, Ph.D., College of Charleston, South Carolina, USA
I am surprised by the apparent absence of eurhinodelphinids – however, I can’t help but notice that NMBE 5023943 looks a bit like one and doesn’t particularly look similar to any named squalodelphinids. In early Miocene assemblages of periotics from South Carolina, there is admittedly a smooth gradient of periotic morphotypes between eurhinodelphinids and waipatiids; indeed, Bianucci et al. (2011) reported some supposed waipatiid periotics that, to me, have some similarities with eurhinodelphinids. Likewise, I would compare both of these figured specimens with the waipatiid periotics from Malta figured by Bianucci et al.
In the conclusion, various morphotypes are mentioned; I could not easily find reference to these in the descriptions, though the description hints at some disparity within the kentriodontid sample. If there are several morphotypes, some more descriptions, comparisons, and commentary is certainly warranted. The different morphotypes ought to be clearly indicated/labeled in the figures.
OMM: Since the current paper is written in English, perhaps SUMM or UMM is a better acronym for the Swiss Upper Marine Molasse.
Lastly, while I appreciate that some paleocetologists prefer not to identify periotics past the family level (e.g. Hiroto Ichishima), I think that perhaps more can be done. Are any of these periotics referable to existing genera, either as “Genus sp.” or “cf. Genus”? For example, it seems as though the authors consider some of the kentriodontid specimens to be referable to Kentriodon proper.
Line 43: perhaps indicate what marine basins (e.g. Mediterranean, Paratethys; others?) are included within the circum-Mediterranean region.
Line 70: Some teeth can be identified to the family level, and occasionally genus level, especially for heterodont dolphins (e.g. Squalodon), and even squalodelphinids and some kentriodontid taxa have distinctive teeth. I suggest some additional attention be given towards teeth, even if not as informative as periotics.
Lines 103-104: what features in particular compare well with Kentriodon spp.?
Line 108: “remains undescribed” (since Kentriodon pernix is singular)
Figures: A figure showing the stratigraphy would be greatly informative, and noting the stratigraphic position of specimens reported within.
Tables: perhaps a table with a faunal list (including additions/revisions made to original faunal list by Pilleri) would be informative.

---

## Round 0.2 · Minor Revisions

Dear authors,

Thank you for your submission to PeerJ. I appreciate your careful revisions based on the previous round of reviews. Both reviewers suggest a few additional revisions, including a few minor additions and justifications to the text. After these revisions are completed, I believe that the manuscript will be publishable in PeerJ.

In addition to the reviewer comments, there were a few suggestions that I had found after reading:

Line 17: ‘micro-computed tomography’ rather than ‘computerized’. Also line 57.

Line 28: Paratethys

Line 110: ‘slightly’

Line 189: Paratethys

Line 201: Paratethys

Line 221–223: Was this large sample disparate in shape or did it have low intraspecific variation?

When you submit your revision, please include a clean copy of the manuscript, a tracked changes version, and a response to reviewers outlining changes that were made.

Thank you for your submission!

Brandon P. Hedrick, Ph.D.

·

Basic reporting

No comment.

Experimental design

No comment.

Validity of the findings

No comment.

Additional comments

The authors have made all of my suggested changes and the manuscript is looking great! I only have some minor suggestions for improvement.

Lines 40-41: wording is a little unclear ‘has become isolated in many odontocetes’ – suggest rephrasing to ‘isolated from the skull’

Line 50: portrays = takes (?)

Line 50: Fig 2 sentence start – suggest rephrase to reference the figure in the context of an explanation, e.g., ‘Families present are xyz (Fig 2)

Fig 2: The figure is great; I love the use of silhouettes. It could take up about half the space, however, i.e., the tree branches could be about half the length that they are since the goal is to show the relationships.

Line 78: ‘annd’ = and

Line 127: a period is missing

Lines 150–152: It would be good to specify which ones have larger lengths

Fig. 7: The tree here could also take up slightly less space if branch lengths are not being shown.

Line 202: ‘th’ = the

Line 260: ‘by the eastern coast’ = ‘on the eastern coast’

·

Basic reporting

The article is still very well-written.

Experimental design

I am still impressed with the approach; I cannot recall if the authors coded their periotic specimens into a matrix in the original manuscript, but this is an excellent - if laborious - method to confirm family level identifications for odontocete periotics - it's one I've considered for my own study of a large assemblage of isolated periotics from the southeastern USA, so I'm pleased to see it being put into place here.

Validity of the findings

I'm still a little skeptical of the assignment of the two periotics to Squalodelphinidae. The authors have pointed out a couple of features - but also note some squalodelphinid features lacking. A long anterior process is present in some squalodelphinids like Notocetus, but also characterizes most eurhinodelphinids; a dorsally opening, and large aperture for the cochlear aqueduct is also present in eurhinodelphinids like Schizodelphis (e.g. Benoit et al., 2011: fig. 5; Geobios 44:323-334). The ventrally deflected anterior process in 5023942 closely resembles eurhinodelphinid periotics from the Lee Creek Mine as well as Xiphiacetus cristatus. Many of the features found in your specimens can be seen in "waipatiid" periotics from Malta as I previously pointed out. Also: what about the trace of the anteroexternal/parabullary sulcus? If memory serves, that's been pointed out as a useful feature to distinguish between eurhinodelphinids, squalodelphinids, and waipatiids by Lambert, Muizon, and Tanaka in the past.

I'm not trying to be nitpicky here; there are loads of strange, difficult-to-identify but at present unpublished specimens that blur the lines between some of these groups - and it would of course be unfair to cite these. I just think the text needs a little more justification in the following specific areas:

1) Note that some of these features show up in other clades (e.g. deep anterior bullar facet, dorsally opening cochlear aqueduct); 2) these periotics are missing some squalodelphinid features (rectangular pars cochlearis; 3) make some brief comparisons with the similar-looking supposed waipatiid periotics reported by Bianucci et al. 2011 from coeval rocks in Malta; and 4) please label the articular rim in figure 5. In 5023942 it looks to be present but small and in 5023943 it is not apparent at all (perhaps damaged like the posterior process, and admittedly not mentioned in the text, perhaps accordingly).

Also, Fordyce 1994 could be cited on line 142 - he also discussed this tubercle in squalodontids.

On the topic of squalodelphinids, the tympanic bulla is in articulation in the S. fabianii holotype and presumably the periotic is still in situ, so perhaps it's more accurate to state that the periotic is unprepared on line 237-238.

Additional comments

I'm very happy with the manuscript overall, and provided that the authors address my specific points about the section on squalodelphinids, I don't think I'll need to see this again as I trust the authors will make these additions.

I apologize if I was not specific enough with my earlier comments/questions!

---

## Round 0.3 · accepted · Accept

Dear authors,

Thank you for your submission to PeerJ and your careful attention to reviewer comments. I now find this manuscript to be publishable and am moving it on to the next stage. Congratulations! Please let me know if you have any additional questions and I would be happy to answer them.

Best,

Brandon P. Hedrick, Ph.D.